# Comparison of machine-learning and logistic regression models for prediction of 30-day unplanned readmission in electronic health records: A development and validation study

**Masao Iwagami**[1,2,3,4☉]*, **Ryota Inokuchi**[1,2,5,6☉], **Eiryo Kawakami**[7,8], **Tomohide Yamada**[9], **Atsushi Goto**[10], **Toshiki Kuno**[11,12], **Yohei Hashimoto**[13,14], **Nobuaki Michihata**[14,15], **Tadahiro Goto**[14,16], **Tomohiro Shinozaki**[17], **Yu Sun**[1,2], **Yuta Taniguchi**[1], **Jun Komiyama**[1,2], **Kazuaki Uda**[1,2], **Toshikazu Abe**[1,18], **Nanako Tamiya**[1,2,3,19]

1 Department of Health Services Research, Institute of Medicine, University of Tsukuba, Tsukuba, Ibaraki, Japan, 2 Health Services Research and Development Center, University of Tsukuba, Tsukuba, Ibaraki, Japan, 3 Digital Society Division, Cyber Medicine Research Center, University of Tsukuba, Tsukuba, Ibaraki, Japan, 4 Faculty of Epidemiology and Population Health, London School of Hygiene and Tropical Medicine, London, United Kingdom, 5 Department of Clinical Engineering, The University of Tokyo Hospital, Tokyo, Japan, 6 Department of Emergency and Critical Care Medicine, The University of Tokyo Hospital, Tokyo, Japan, 7 Department of Artificial Intelligence Medicine, Graduate School of Medicine, Chiba University, Chiba, Chiba, Japan, 8 Advanced Data Science Project (ADSP), RIKEN Information R&D and Strategy Headquarters, RIKEN, Yokohama, Kanagawa, Japan, 9 Department of Diabetes and Metabolic Diseases, Graduate School of Medicine, The University of Tokyo, Tokyo, Japan, 10 Department of Public Health, School of Medicine, Yokohama City University, Yokohama, Kanagawa, Japan, 11 Division of Cardiology, Montefiore Medical Center, Albert Einstein College of Medicine, NY, United States of America, 12 Cardiology Division, Massachusetts General Hospital, Harvard Medical School, Boston, MA, United States of America, 13 Department of Ophthalmology, Graduate School of Medicine, The University of Tokyo, Tokyo, Japan, 14 Department of Clinical Epidemiology and Health Economics, School of Public Health, The University of Tokyo, Tokyo, Japan, 15 Cancer Prevention Center, Chiba Cancer Center Research Institute, Chiba, Japan, 16 TXP Medical Co. Ltd, Tokyo, Japan, 17 Department of Information and Computer Technology, Faculty of Engineering, Tokyo University of Science, Tokyo, Japan, 18 Department of Emergency and Critical Care Medicine, Tsukuba Memorial Hospital, Tsukuba, Ibaraki, Japan, 19 Center for Artificial Intelligence Research, University of Tsukuba, Tsukuba, Ibaraki, Japan

☉ These authors contributed equally to this work.
* masao.iwagami@lshtm.ac.uk

**Data Availability Statement:** Restrictions apply to the availability of these data, which were used

## Abstract

It is expected but unknown whether machine-learning models can outperform regression models, such as a logistic regression (LR) model, especially when the number and types of predictor variables increase in electronic health records (EHRs). We aimed to compare the predictive performance of gradient-boosted decision tree (GBDT), random forest (RF), deep neural network (DNN), and LR with the least absolute shrinkage and selection operator (LR-LASSO) for unplanned readmission. We used EHRs of patients discharged alive from 38 hospitals in 2015–2017 for derivation and in 2018 for validation, including basic characteristics, diagnosis, surgery, procedure, and drug codes, and blood-test results. The outcome was 30-day unplanned readmission. We created six patterns of data tables having different numbers of binary variables (that ≥5% or ≥1% of patients or ≥10 patients had) with and without blood-test results. For each pattern of data tables, we used the derivation data to

under license for the current study, and so are not publicly available. The data that support the findings of this study are available from Medical Data Vision Co., Ltd. (MDV) through a data request application process (https://en.mdv.co.jp). Researchers can contact the company for more information.

**Funding:** This study was supported by the Cyber Medicine Research Center, University of Tsukuba, Tsukuba, Ibaraki, Japan, and a Japan Society for the Promotion of Science (JSPS) KAKENHI Grant (No. 19K19430) from the Japanese Ministry of Education, Culture, Sports, Science, and Technology. The funders had no role in study design, data collection, data analysis, data interpretation, or writing.

**Competing interests:** The authors have declared that no competing interests exist.

establish the machine-learning and LR models, and used the validation data to evaluate the performance of each model. The incidence of outcome was 6.8% (23,108/339,513 discharges) and 6.4% (7,507/118,074 discharges) in the derivation and validation datasets, respectively. For the first data table with the smallest number of variables (102 variables that ≥5% of patients had, without blood-test results), the c-statistic was highest for GBDT (0.740), followed by RF (0.734), LR-LASSO (0.720), and DNN (0.664). For the last data table with the largest number of variables (1543 variables that ≥10 patients had, including blood-test results), the c-statistic was highest for GBDT (0.764), followed by LR-LASSO (0.755), RF (0.751), and DNN (0.720), suggesting that the difference between GBDT and LR-LASSO was small and their 95% confidence intervals overlapped. In conclusion, GBDT generally outperformed LR-LASSO to predict unplanned readmission, but the difference of c-statistic became smaller as the number of variables was increased and blood-test results were used.

## Author summary

It has been controversial over whether machine-learning models can outperform traditional statistical models, such as a logistic regression (LR) model, for the prediction of hospital readmission in electronic health records (EHRs). Therefore, this study aimed to systematically compare the predictive performance of the 30-day unplanned readmission among several machine-learning models and a LR model. We created 6 patterns of data tables according to the number of binary predictor variables (that ≥5% or ≥1% of patients, or ≥10 patients had) with and without blood-test results, expecting that some machine-learning models may outperform the LR model more prominently if the data become richer. We found that the gradient-boosting decision tree (one of machine-learning models) generally outperformed the LR model. However, against our expectation, the difference in the predictive performance between them was smaller in the last data table with the largest number of variables (1543 variables including blood-test results). Thus, this study concludes that the superiority of machine-learning methods to traditional statistical models may not be larger in EHRs with richer information. Future studies should focus on other potential predictors in EHRs, such as images and processed natural language, for demonstrating the superior performance of machine-learning methods to traditional statistical models.

## Introduction

Unplanned hospital readmission is a common issue in public health because it imposes burdens on medical budgets, healthcare staff, and patients [1]. Some unplanned hospital readmission is preventable [2]. Therefore, it is important to identify patients with a high risk of unplanned readmission for intervention. For achieving this, a clinical prediction model that can accurately estimate the probabilities of clinical outcomes for individual patients is essential [3].

Clinical prediction models have been developed for 30- or 28-day unplanned hospital readmission—mostly logistic regression (LR) models based on a small number of clinical variables [4,5]. However, the prediction ability of these models is often suboptimal, with a c-statistic or

area under the receiver operating characteristic curve (AUROC) of <0.70. It was discussed that the prediction ability should be improved by increasing the number/variety of clinical variables and developing more accurate statistical or mathematical models with modern computing techniques [4,5].

Recently, machine-learning models, such as the gradient-boosted decision tree (GBDT), random forest (RF), and deep neural network (DNN), have been applied to claims data and electronic health records (EHRs) to build prediction models for readmission [6–8]. However, according to a recent systematic review, there was no statistically significant difference in the predictive performance between "traditional regression models" (defined in the paper as LR or Cox regression models, regardless of feature selection techniques) and machine-learning models, with average c-statistic values of 0.71 for 26 studies and 0.74 for 15 studies, respectively, corresponding to a difference in c-statistic of 0.03 (95% confidence interval [CI]: –0.01 to 0.07) [8]. As its major limitation, however, only few studies included in the systematic review involved direct comparisons of "traditional regression models" and machine-learning models within the same study [9–13]. Moreover, the studies included in the systematic review tended to use a small number of predictor variables. The predictive performance of machine-learning models may depend on the number of predictor variables, as well as the inclusion of continuous variables potentially having nonlinear relationships with a study outcome, such as blood-test results. Some machine-learning models may outperform an LR model if the number of predictor variables is increased and blood-test results are used for prediction.

Thus, using the EHRs obtained from 38 Japanese hospitals, we aimed to develop, validate, and compare GBDT, RF, DNN, and LR with the least absolute shrinkage and selection operator (LR-LASSO) for predicting 30-day unplanned readmission, by intentionally creating six patterns of data tables having different numbers of predictor variables (which ≥5% or ≥1% of patients or ≥10 patients had) with and without blood-test results. Notably, among several feature-selection techniques for LR, such as the stepwise variable selection and the LASSO, we decided to use the LASSO because it is generally less likely to cause overfitting than other techniques [14] and also because LR-LASSO outperformed LR with the stepwise variable selection in a study predicting 30-day all-cause non-elective readmission [11]. Our study hypothesis was that some machine-learning models (GBDT, RF, or DNN) outperform LR-LASSO more prominently if the number of predictor variables is increased and blood-test results are used. The findings of this study could help hospitals apply clinical prediction models to their EHRs to identify patients at high risk for readmission and intervene efficiently.

## Results

### Characteristics of study participants

In the Medical Data Vision (MDV) database, we identified 635,509 discharges of 410,941 patients, with at least one blood test during hospitalization, who were admitted after January 1, 2015 and discharged before December 31, 2018, from 38 hospitals (**Fig 1**). After the exclusion of admissions associated with childbirth, discharges dead, and transfers to other hospitals and cases of missing data, there were 457,587 discharges eligible for analysis, including 339,513 discharges (mean age 62.0 ± 24.6 years, 54.3% men) from January 1, 2015 to December 31, 2017 in the derivation dataset and 118,074 discharges (mean age 63.4 ± 24.1 years, 54.1% men) from January 1 to December 31, 2018 in the validation dataset.

The incidence of 30-day unplanned readmission was 6.8% (23,108/339,513) and 6.4% (7,507/118,074) for the derivation and validation datasets, respectively. **Table 1** presents the basic characteristics of the patients overall and by outcome status. Patients with 30-day unplanned readmission tended to be older men and were more likely to have been hospitalized

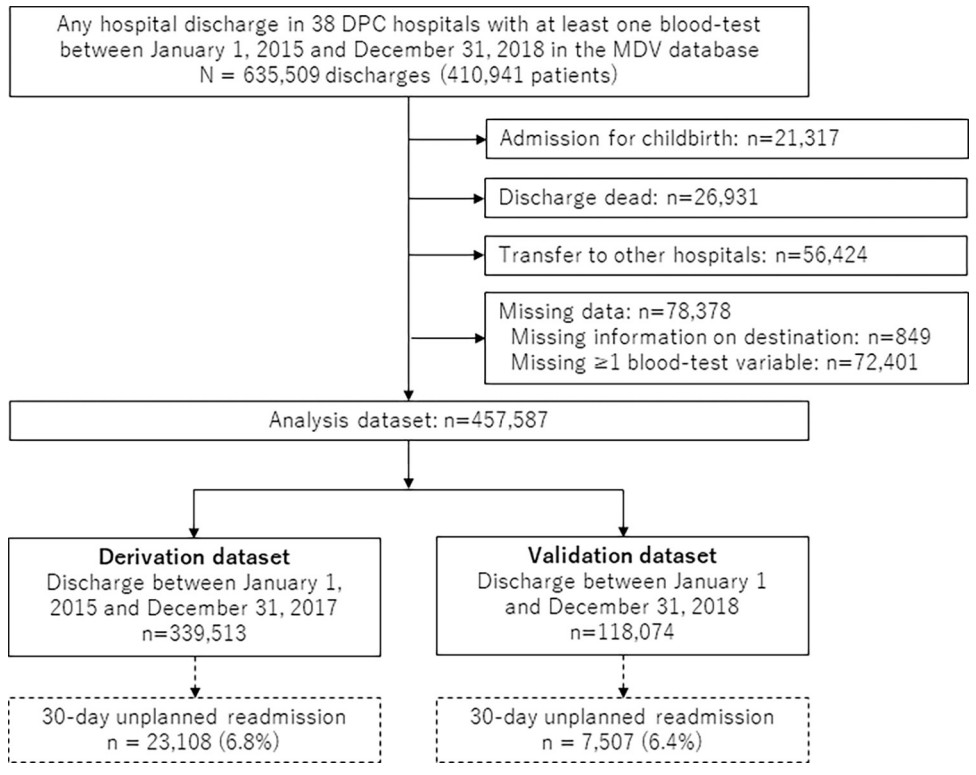

**Fig 1. Flowchart.** DPC, Diagnosis Procedure Combination; MDV, Medical Data Vision.

in the past year, admitted for diseases of the respiratory or digestive system, and discharged to nursing homes than those without 30-day unplanned readmission.

## Characteristics of created datasets

Among the 2102 International Classification Disease 10th revision (ICD-10) codes, 914 surgery codes, 122 procedure codes, and 468 Anatomical Therapeutic Chemical (ATC) codes, we excluded variables that <10 patients had, in both the derivation and validation datasets, to avoid the "perfect separation" (or "complete separation") problem [15]. Consequently, 823 ICD-10 codes (**S1 Table**), 259 surgery codes (**S2 Table**), 65 procedure codes (**S3 Table**), and 381 ATC codes (**S4 Table**) were regarded as candidate binary predictor variables for the following analyses. The distribution of the 10 types of blood-test results, as the last measurement during hospitalization, is presented in **S5 Table**.

Then, we created six pattens of data tables with different numbers of the candidate binary predictor variables (that ≥5% or ≥1% of patients or ≥10 patients had, in both the derivation and validation datasets) with and without blood-test results, in addition to the basic characteristics presented in **Table 1**. The number of candidate predictor variables was 102 for patten 1 (including binary variables that ≥5% had, without blood-test results), 112 for pattern 2 (including binary variables that ≥5% had, with blood-test results), 296 for patten 3 (including binary variables that ≥1% had, without blood-test results), 306 for pattern 4 (including binary variables that ≥1% had, with blood-test results), 1533 for pattern 5 (including binary variables that ≥10 patients had, without blood-test results), and 1543 for pattern 6 (including binary variables that ≥10 patients had, with blood-test results).

**Table 1. Baseline characteristics of patients.**

| | Derivation dataset (N = 339,513) | | | Validation dataset (N = 118,074) | | |
|---|---|---|---|---|---|---|
| | Total N = 339,513 | Outcome (-) N = 316,405 | Outcome (+) N = 23,108 | Total N = 118,074 | Outcome (-) N = 110,567 | Outcome (+) N = 7,507 |
| Age, mean±SD | 62.0±24.6 | 61.5±24.7 | 68.4±22.6 | 63.4±24.1 | 63.0±24.2 | 69.0±22.3 |
| Sex (men), n (%) | 184,218 (54.3) | 170,962 (54.0) | 13,256 (57.4) | 63,918 (54.1) | 59,607 (53.9) | 4,311 (57.4) |
| No. of hospitalization in the past year, n (%) | | | | | | |
| 0 | 209,597 (61.7) | 200,083 (63.2) | 9,514 (41.2) | 73,585 (62.3) | 70,487 (63.8) | 3,098 (41.3) |
| 1 | 65,259 (19.2) | 60,075 (19.0) | 5,184 (22.4) | 23,147 (19.6) | 21,379 (19.3) | 1,768 (23.6) |
| ≧2 | 64,657 (19.0) | 56,247 (17.8) | 8,410 (36.4) | 21,342 (18.1) | 18,701 (16.9) | 2,641 (35.2) |
| Admission diagnosis category (ICD-10 code), n (%) | | | | | | |
| Certain infectious and parasitic diseases (A00-B99) | 10,555 (3.1) | 9,865 (3.1) | 690 (3.0) | 3,398 (2.9) | 3,211 (2.9) | 187 (2.5) |
| Neoplasms (C00-D48) | 92,994 (27.4) | 86,579 (27.4) | 6,415 (27.8) | 30,828 (26.1) | 28,761 (26.0) | 2,067 (27.5) |
| Diseases of the blood, blood-forming organs and immune mechanism (D50-D89) | 3,525 (1.0) | 3,086 (1.0) | 439 (1.9) | 1,113 (0.9) | 960 (0.9) | 153 (2.0) |
| Endocrine, nutritional and metabolic diseases (E00-E90) | 11,117 (3.3) | 10,309 (3.3) | 808 (3.5) | 3,693 (3.1) | 3,438 (3.1) | 255 (3.4) |
| Mental and behavioral disorders (F00-F99) | 402 (0.1) | 382 (0.1) | 20 (0.1) | 152 (0.1) | 148 (0.1) | 4 (0.1) |
| Diseases of the nervous system (G00-G99) | 6,153 (1.8) | 5,811 (1.8) | 342 (1.5) | 2,313 (2.0) | 2,179 (2.0) | 134 (1.8) |
| Diseases of the eye and adnexa (H00-H59) | 741 (0.2) | 709 (0.2) | 32 (0.1) | 256 (0.2) | 250 (0.2) | 6 (0.1) |
| Diseases of the ear and mastoid process (H60-H95) | 3,481 (1.0) | 3,353 (1.1) | 128 (0.6) | 1,209 (1.0) | 1,161 (1.1) | 48 (0.6) |
| Diseases of the circulatory system (I00-I99) | 48,061 (14.2) | 45,330 (14.3) | 2,731 (11.8) | 17,603 (14.9) | 16,650 (15.1) | 953 (12.7) |
| Diseases of the respiratory system (J00-J99) | 45,224 (13.3) | 41,191 (13.0) | 4,033 (17.5) | 15,389 (13.0) | 14,095 (12.7) | 1,294 (17.2) |
| Diseases of the digestive system (K00-K93) | 47,648 (14.0) | 43,846 (13.9) | 3,802 (16.5) | 17,380 (14.7) | 16,136 (14.6) | 1,244 (16.6) |
| Diseases of the skin and subcutaneous tissue (L00-L99) | 4,233 (1.2) | 3,954 (1.2) | 279 (1.2) | 1,471 (1.2) | 1,377 (1.2) | 94 (1.3) |
| Diseases of the musculoskeletal system and connective tissue (M00-M99) | 13,387 (3.9) | 12,945 (4.1) | 442 (1.9) | 5,041 (4.3) | 4,888 (4.4) | 153 (2.0) |
| Diseases of the genitourinary system (N00-N99) | 19,440 (5.7) | 18,078 (5.7) | 1,362 (5.9) | 7,397 (6.3) | 6,899 (6.2) | 498 (6.6) |
| Symptoms, signs and abnormal clinical and laboratory findings (R00-R99) | 6,516 (1.9) | 5,982 (1.9) | 534 (2.3) | 1,826 (1.5) | 1,708 (1.5) | 118 (1.6) |
| Injury, poisoning and certain other consequences of external causes (S00-T98) | 26,036 (7.7) | 24,985 (7.9) | 1,051 (4.5) | 9,005 (7.6) | 8,706 (7.9) | 299 (4.0) |
| Discharge place (discharge to nursing home [vs. home]), n (%) | 19,214 (5.7) | 16,917 (5.3) | 2,297 (9.9) | 6,758 (5.7) | 6,095 (5.5) | 663 (8.8) |

Abbreviations: SD, standard deviation; ICD-10, International Classification of Diseases 10th Revision.

## Comparison of performances between prediction models

For each pattern, using the derivation dataset, each model was developed with optimized hyperparameters (**S6 Table**).

By applying these models to the validation dataset, the discrimination ability of each model —indicated by the c-statistic or AUROC—was evaluated, as shown in **Fig 2** and **S7 Table**. The GBDT outperformed LR-LASSO for all patterns (Delong's test P-value <0.001). The discrimination ability of DNN was the worst for all patterns. In more detail, for pattern 1 (including binary variables that ≥5% had, without blood-test results), which had the smallest number of variables, the point estimate in c-statistic was the highest for GBDT (0.740), followed by RF (0.734), LR-LASSO (0.720), and DNN (0.664). For pattern 6 (including binary variables that

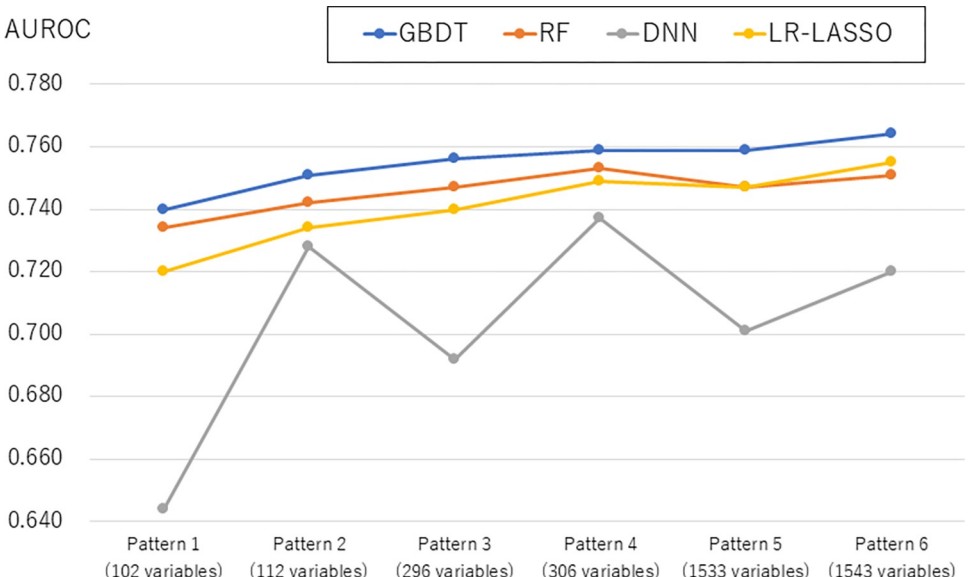

**Fig 2. C-statistic of each model for six patterns of data tables having different numbers of the candidate predictor variables with and without blood-test results.** AUROC, area under the receiver operating characteristic curve; CI, confidence interval; GBDT, gradient-boosted decision tree; RF, random forest; DNN, deep neural network; LR-LASSO, logistic regression with the least absolute shrinkage and selection operator. Pattern 1: Basic characteristics and binary predictor variables that $\geq$5% of patients had, without blood-test results (102 variables). Pattern 2: Basic characteristics and binary predictor variables that $\geq$5% of patients had, without blood-test results (112 variables). Pattern 3: Basic characteristics and binary predictor variables that $\geq$1% of patients had, without blood-test results (296 variables). Pattern 4: Basic characteristics and binary predictor variables that $\geq$1% of patients had, with blood-test results (306 variables). Pattern 5: Basic characteristics and binary predictor variables that $\geq$10 patients had, without blood-test results (1533 variables). Pattern 6: Basic characteristics and binary predictor variables that $\geq$10 patients had, with blood-test results (1543 variables).

$\geq$10 patients had, with blood-test results), which had the largest number of variables, the point estimate in c-statistic was highest for GBDT (0.764), followed by LR-LASSO (0.755), RF (0.751), and DNN (0.720), suggesting that the difference between GBDT and LR-LASSO was small. The 95% confidence intervals of GBDT (0.758–0.769) and LR-LASSO (0.749–0.761) overlapped.

We also evaluated the goodness of fit indicators, including the $R^2$ Cox–Snell, $R^2$ Nagelkerke, and Brier score (**S8 Table**). The overall trend was broadly similar to that of the c-statistic. GBDT was similar or only slightly better, with higher $R^2$ scores and a lower Brier score, than those of LR-LASSO in every pattern. Regarding the indicators for calibration (**S9 Table**), the calibration-in-the-large of GBDT tended to be better (smaller) than that of LR-LASSO in every model, although the calibration slope of LR-LASSO tended to be better (closer to 1) than that of GBDT.

## Details of the prediction models in the data table with the largest number of variables

For pattern 6 (including binary variables that $\geq$10 patients had, with blood-test results), the ROC curves are shown in **Fig 3**, and the calibration plots are shown in **Fig 4**. As graphically shown, the calibration of every model was similarly good for patients at low risk for readmission. However, GBDT and LR-LASSO tended to overestimate, whereas RF and DNN tended to underestimate the probability of readmission for patients at higher risk for readmission.

For pattern 6 (including binary variables that $\geq$10 patients had, with blood-test results), **S10 Table** presents regression coefficients in LR-LASSO, whereas **Fig 5** shows the variable

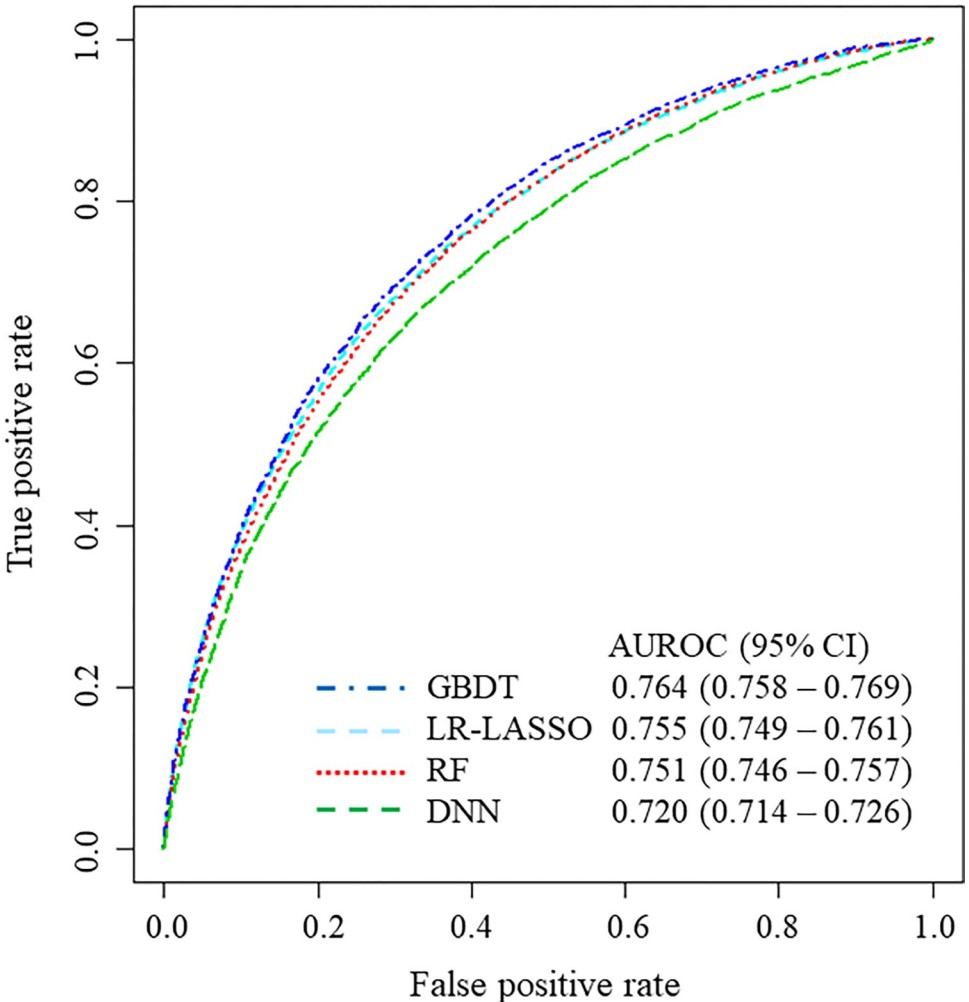

**Fig 3. Receiver operating characteristic curve of each model for the data table with the largest number of variables including blood-test results (1543 variables).** AUROC, area under the receiver operating characteristic curve; CI, confidence interval; GBDT, gradient-boosted decision tree; RF, random forest; DNN, deep neural network; LR-LASSO, logistic regression with the least absolute shrinkage and selection operator.

importance of the top 10 important predictors for GBDT, suggesting that age, blood-test results, and number of hospitalizations in the past year were important predictors. By excluding each of the 10 variables, the c-statistic tended to be reduced, suggesting that these variables are important predictors (**S11 Table**).

## Discussion

Using the EHRs of 38 Japanese hospitals, we systematically compared the prediction performance for 30-day unplanned readmission among commonly used machine-learning models (GBDT, RF, and DNN) and an LR-LASSO model, by creating six patterns of data tables having different numbers of predictor variables (that ≥5% or ≥1% of patients or ≥10 patients had) with and without blood-test results. The discrimination ability mostly improved if the number of predictor variables was increased or blood-test results were added. For the latter patterns of data tables, the c-statistics of the models except for DNN were approximately 0.75 or higher, suggesting that the performance of these models was better than or comparable to that of

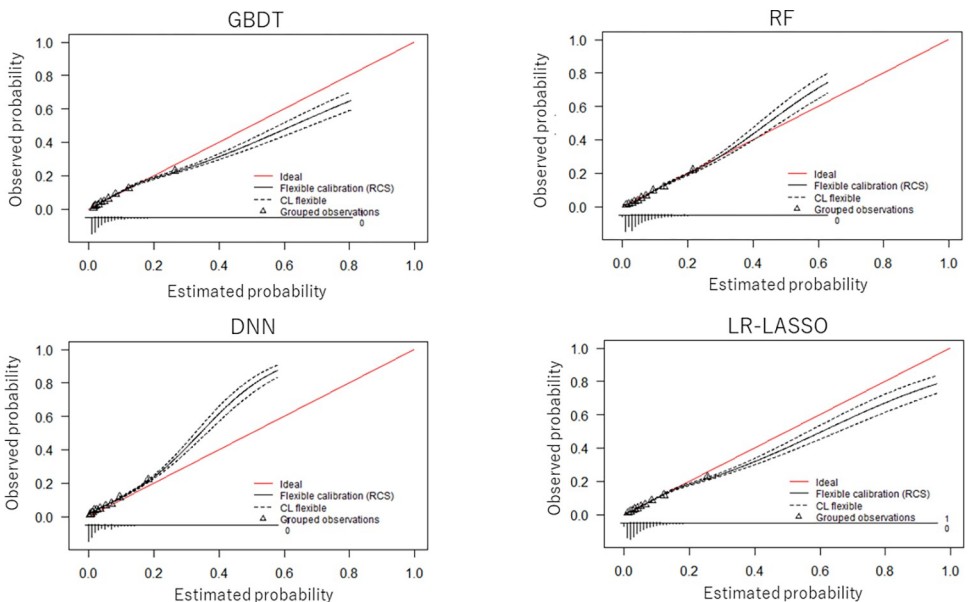

**Fig 4. Calibration plot of each model for the data table with the largest number of variables including blood-test results (1543 variables).** GBDT, gradient-boosted decision tree; RF, random forest; DNN, deep neural network; LR-LASSO, logistic regression with the least absolute shrinkage and selection operator.

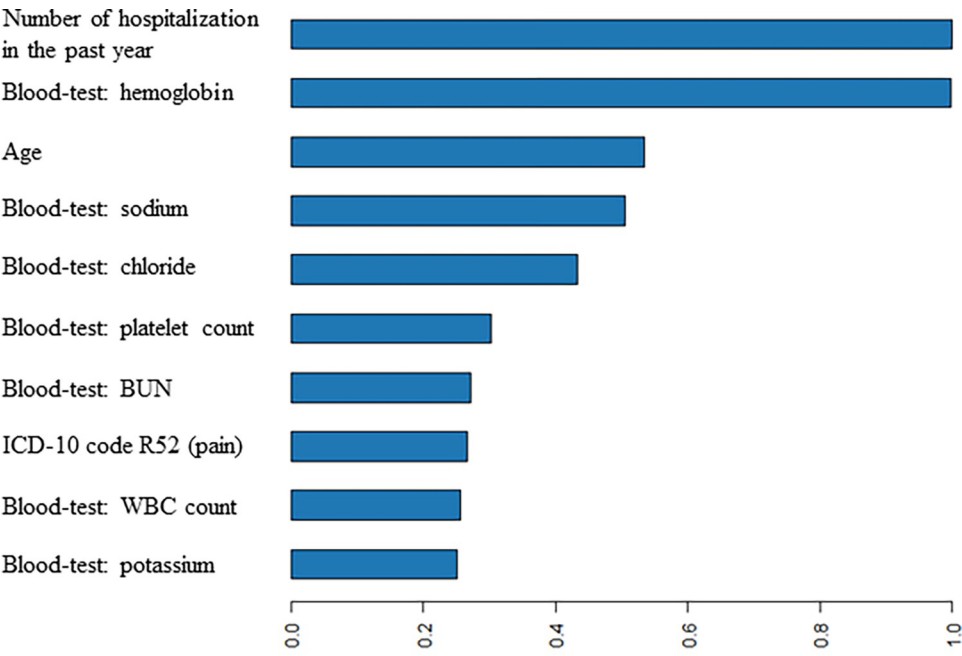

**Fig 5. Variable importance of the top 10 important predictors in gradient-boosted decision tree for the data table with the largest number of variables including blood-test results (1543 variables).** ICD-10, International Classification Disease 10th revision; BUN, blood urea nitrogen; WBC, white blood cell.

previously reported models [4,5,8]. For each pattern of data table, the c-statistic of GBDT was higher than that of LR-LASSO. However, against our hypothesis (that some machine-learning models outperform LR-LASSO more prominently if the number of predictor variables is increased and blood-test results are used), the difference in c-statistics between GBDT and LR-LASSO became rather smaller in the last pattern of data table, which had the largest number of binary variables and blood-test results.

To our knowledge, this was the first systematic comparison of the predictive performance of commonly used machine-learning models and an LR model for unplanned readmission, using data tables characterized by different numbers of predictor variables and the presence/absence of blood-test results. A recent systematic review [8] included several studies comparing the performance of different prediction models for readmission with the same dataset; however, the target population was narrow, e.g., patients with lupus [12] and heart failure [13]. Our study is partly similar to that of Jamei et al. [9], who used the EHRs of >300,000 patients in Northern California, USA to compare the predictive performance of RF with 100 features; NN with 100, 500, and 1667 features; and LR with 100, 500, and 1667 features. With 100 features, RF and NN outperformed LR, with AUROCs of 0.77, 0.76, and 0.72, respectively. With 1667 features, NN performed far better than LR, with AUROCs of 0.78 and 0.66, respectively. We speculate that the reduction in the predictive performance of LR according to an increase in the number of features was probably due to overfitting, as it appears that the authors did not conduct any regularization or penalization via means such as LASSO, which we used in the present study. In contrast to their study, in the present study, the predictive performance of DNN was worse than that of the other models. This is probably because our dataset, which included mostly binary variables and only a small number of continuous variables (such as age and 10 blood-test results), was simpler than the dataset of Jamei et al., which had a larger number of continuous variables, such as vital signs and 26 blood-test results [9]. A recent meta-analysis that compared the performances of advanced machine learning algorithms and logistic regression in predicting hospital readmissions concluded that DNN performed the best [16], suggesting that the potential of DNN should be further explored.

To date, comparisons of machine-learning models and "traditional regression models" (defined as LR or Cox regression models, regardless of feature selection techniques [8]) have been performed for various outcomes. Christodoulou et al. conducted a systematic review of 71 studies on clinical prediction models, regardless of outcomes, and concluded that there was no evidence of machine-learning methods outperforming LR [17]. However, the systematic review was limited in that it included only studies with <100 predictor variables; the authors identified seven studies with >100 predictor variables, all of which were excluded owing to "high risk of bias" in their classification [17]. There have been several published studies after the systematic review, and these studies seem to have similar limitations [18–20]. Therefore, it remained unclear whether machine-learning methods can outperform LR if the dataset includes a larger number of predictor variables. Our study answers this question: we intentionally created six patterns of data tables having different numbers of predictor variables (102, 112, 296, 306, 1533, or 1543) with and without blood-test results, and in the data table with the largest number of variables, the difference in c-statistics between GBDT and LR was rather small. Therefore, the number of predictor variables does not seem to affect the results and conclusions very much when comparing the predictive performance of the machine-learning and LR models. Instead, the types or characteristics of predictors may matter. Notably, in the present study, the vast majority of the predictor variables were binary, and the number of continuous variables was small. An increased number of continuous variables, particularly those having nonlinear relationships with a study outcome, and the use of more complex information (such as images and processed natural language) may be needed to demonstrate the

superiority of machine-learning methods over "traditional regression models" in clinical risk prediction. Further studies are needed to clarify this aspect.

Regarding practical implications, according to the results of the present study, the GBDT may be recommended for hospitals applying clinical prediction models to their EHRs to identify patients at high risk for readmission. However, the benefit of GBDT over LR-LASSO was small, suggesting that "traditional regression models" such as an LR model may also be acceptable, considering that they are more user-friendly and easier to interpret. In GBDT, the variable importance was high for age, blood-test results, and number of hospitalizations in the previous year, which also decreased the c-statistic when they were excluded from the model. Therefore, clinicians and clinical staff should be conscious of these variables when their patients are to be discharged. Patients at high risk for readmission may need to have their discharge postponed or receive intensive post-discharge follow-up as an outpatient [2] or via remote monitoring with a smartphone or wearable device at home [21].

This study had several limitations. First, we used data from a relatively large number of hospitals, but it is unclear whether these hospitals are representative of Japan. Although we believe that the selection of these hospitals is unlikely to be associated with the performance of the machine-learning models, our findings should be validated using data from different hospitals in Japan as well as hospitals in other countries. Second, we were unable to differentiate between patients admitted to different hospitals, because the hospital identifier was omitted to preserve privacy and data safety in the MDV database. Therefore, we could not account for any clustering effect by hospitals in our analysis. Third, the incidence of 30-day unplanned readmission was meaningfully lower in 2018 in the validation dataset (6.4%) than that in 2015–2017 in the derivation dataset (6.8%), possibly suggesting a decreasing trend in hospital readmission in Japan. This trend might have harmed the applicability of the developed models to the validation dataset, resulting in underestimation of the predictive ability of the models. Fourth, although the MDV database contained important information on recorded diagnoses and treatments during hospitalization, as well as blood-test results, it lacks other clinical variables that may be useful for the prediction of hospital readmission, such as vital signs and socioeconomic features [8]. Finally, we used a hospital-based database and could only predict readmission to the hospital that the patient was initially admitted to, rather than readmission to any hospital. In emergencies, patients may be sent to another hospital for readmission. Thus, a study utilizing a population-based database should be conducted to assess the performance of machine-learning methods in predicting readmission to any hospital.

## Conclusions

Using the EHRs of 38 hospitals in the MDV database, we compared the predictive performance of machine-learning and LR models for six patterns of data tables having different numbers of predictor variables with and without blood-test results. For every pattern, GBDT outperformed LR-LASSO. However, the difference in c-statistic between GBDT and LR-LASSO was small, especially in the data table with the largest number of variables (1543 variables including blood-test results). Thus, instead of the number of predictor variables, future studies should focus on other potential predictor variables in EHRs, such as images and processed natural language, for demonstrating the superior performance of machine-learning methods to "traditional regression models".

## Materials and methods

### Design and setting

We conducted a retrospective cohort study using a hospital-based database in Japan.

## Data source

Under the universal health care system in Japan, Diagnosis Procedure Combination (DPC) data are administrative claims data and discharge summaries that are collected at the top of the DPC system, a case-mix patient classification, and a lump-sum payment system for inpatients in acute care hospitals [22]. The MDV database, which was built by Medical Data Vision Co., Ltd. (Tokyo, Japan), consisted of >350 acute care hospitals—approximately 20% of the DPC hospitals in Japan. It consisted of acute care hospitals that used the business support system of MDV Co., Ltd. and provided consent for secondary data use for research purposes. The age distribution of inpatients in the MDV database is similar to that of all DPC hospitals, whereas the hospital volume (i.e., number of beds) tends to be larger [23]. During the study period between January 1, 2015 and January 31, 2019, laboratory test values were available from 38 hospitals that agreed to provide the data. Thus, in this study, we used DPC data (administrative claims data and discharge summaries) and blood-test results from 38 hospitals. We did not conduct any sample size calculation, but planned to use all the available data.

The DPC data included basic patient characteristics such as age and sex; recorded diagnoses based on ICD-10 codes [24], including the admission diagnosis (indicating the reason for hospital admission), main diagnosis, most resource-consuming diagnosis, second-most resource-consuming diagnosis, comorbidities present on admission (up to four diagnoses), and complications arising after admission (up to four diagnoses), recorded by the responsible physician at the time of discharge; Japanese original surgery codes [25], if a patient received surgery during hospitalization; Japanese original procedure codes [26], such as mechanical ventilation and dialysis; and original Japanese prescription codes (including both oral and intravenous drugs), which are linked to the ATC codes [27]. In addition, the admission status (planned or unplanned) and discharge status (dead, discharged to nursing facility or home, transferred to other hospitals) are available.

The study was approved by the Ethics Committee of the University of Tsukuba (approval no. 1414) in accordance with the Declaration of Helsinki. Because the claims data were anonymized before the researchers received them, the committee approved that individual participants' informed consent was waived according to the ethical guidelines for medical and health research involving human subjects [28]. To ensure privacy and data safety, the hospital identifier was omitted by MDV Co., Ltd. in the data that we received. Thus, we could not identify the names and characteristics of the 38 hospitals and did not know which of the 38 hospitals the patients were admitted to.

## Study participants

We identified the discharges of patients who were admitted to the 38 hospitals, with at least one blood-test during hospitalization, from January 1, 2015 to December 31, 2018. We excluded: i) cases with the admission diagnosis category suggesting childbirth (ICD-10 O00-Q99), ii) discharges dead, iii) transfers to other hospitals, and iv) cases with missing values of discharge location, or one or more of 10 basic blood-tests (white cell count, hemoglobin, platelet count, sodium, potassium, chloride, creatinine, blood urea nitrogen, aspartate transaminase, and alanine transaminase). We excluded cases with missing values of one or more of 10 basic blood-tests because (i) missingness is not likely to be at random, and (ii) even if missingness is at random, strategies to impute missing data in different machine-learning models are different or not established [29], whereas the objective of the present study was to compare the predictive performance of different models for the same complete dataset. If the same patient was hospitalized multiple times during the study period, we considered each admission

as an independent admission, while we used the number of hospitalizations in the past year as a predictor variable to reduce dependence between them.

The data were split into derivation data for patients discharged from January 1, 2015 to December 31, 2017, which were used to develop the models, and validation data for patients discharged from January 1 to December 31, 2018, which were used to assess the discrimination ability and calibration of the models.

## Outcome

The outcome of interest was 30-day unplanned readmission to the hospital from which the patient was initially discharged.

## Predictor variables and data tables

The predictor variables used in this study consisted of basic characteristics, recorded diagnoses, inpatient treatments, and blood-test results that were the last measurement before discharge. The basic characteristics included age (as a continuous variable), sex, admission diagnosis category (according to the ICD-10 codes, from A to T), number of hospitalizations in the past year, and discharge location (home or nursing home). For recorded diagnoses, information on ICD-10 codes recorded anywhere in the DPC code position (i.e., admission, main, most or second-most resource-consuming, comorbidities present on admission, or complications arising after admission) was transformed into the presence or absence of each 3-digit ICD-10 code from A00 to Z00. In total, 2102 types of 3-digit ICD-10 codes in the WHO ICD-10 Version 2010 were used in this study [24]. Similarly, for inpatient treatments, information on the surgery, procedure, and prescribed drugs recorded in the DPC system was transformed into the presence or absence of 914 surgery codes from K000 to K939 [25], 122 procedure codes from J000 to J129 [26], and 468 ATC codes from A01A0 to V07A0 [27].

Among the 2102 ICD-10 codes, 914 surgery codes, 122 procedure codes, and 468 ATC codes, we excluded variables that <10 patients had, in both the derivation and validation datasets, to avoid the "perfect separation" (or "complete separation") problem [15]. Then, using these candidate binary predictor variables, we created six patterns of data tables having different numbers of the variables (that ≥5% or ≥1% of patients or ≥10 patients, in both the derivation and validation datasets) with and without blood-test results, in addition to the basic characteristics (i.e., age, sex, admission diagnosis category, number of hospitalizations in the past year, and discharge location). The six patterns of data tables only differ in the number of predictor variables; the included patients (i.e., 339,513 patients for derivation and 118,074 patients for validation) were the same in every data table.

The continuous variables, including age, the number of hospitalizations in the past year, and blood test results, were standardized to a mean of 0 and a standard deviation of 1. The categorical variables, including the number of hospitalizations in the past year and admission diagnosis category (16 values from A to T), were transformed into a combination of binary variables using dummy values.

## Machine-learning and LR models

Details regarding the supervised machine-learning models are presented elsewhere [15,30]. In brief, RF comprises ensembles of decision trees constructed from bootstrapped training samples, for which random samples corresponding to specific numbers of predictors are selected to initiate tree induction. The GBDT is another ensemble method, in which new tree models for predicting the errors and residuals of previous models are constructed in sequence. These new models are combined, and a gradient descent algorithm is used to minimize the loss

function. A DNN comprises multiple processing layers and model outcomes via intermediate hidden units, each comprising a linear combination of predictors that are transformed into nonlinear functions.

There are several feature-selection techniques for LR, such as the stepwise variable selection and the LASSO [3]. In this study, we used the LASSO because it is generally less likely to cause overfitting than other techniques [14]. Additionally, in a previous study, LR-LASSO outperformed LR with the stepwise variable selection in predicting 30-day all-cause non-elective readmission [11]. Whether the LR-LASSO is a machine-learning or "traditional regression model" may be controversial. However, in accordance with previous systematic reviews comparing machine-learning methods and LR [8,17], we regarded the LR-LASSO as a "traditional regression model". We did not include any interaction terms between the features in our LR-LASSO.

## Statistical analysis

First, we described the distribution or proportions of all the predictor variables according to outcome status in the derivation and validation data. Next, for each pattern of data tables from 1 to 6, we used derivation data to establish the machine-learning and LR models. The hyperparameters for each model were determined and optimized via 10-fold cross-validation within the derivation data (i.e., training data) [31], using automated machine learning in the *h2o* package of R.

Then, using the validation data (i.e., test data), we evaluated the performance of each model with the indicators for discrimination (c-statistic or AUROC), overall fit ($R^2$ Nagelkerke, $R^2$ Cox–Snell, and Brier score), and calibration (calibration plots, calibration-in-the-large, observed/expected, calibration slope, and integrated calibration index). We conducted Delong's tests to compare the c-statistics of machine-learning models and LR-LASSO as a reference.

By focusing on GBDT, which showed the best performance among the studied models, we identified the 10 variables with the highest importance to examine the contribution of each predictor to the model with the best discriminatory abilities [32]. In addition, we demonstrated the change in the c-statistic by excluding each of the 10 variables.

Data cleaning was conducted using STATA version 16 (StataCorp LLC, Texas, USA), and a statistical analysis was performed using R version 4.1.2 (R Foundation for Statistical Computing, Vienna, Austria) with the *h2o* and *rms* packages.

We followed the checklist of the Transparent Reporting of a multivariable prediction model for Individual Prognosis or Diagnosis (TRIPOD) Statement [33].

## Supporting information

**S1 Table. Details and frequency of diagnosis codes (International Classification of Diseases 10th Revision codes).**
(DOCX)

**S2 Table. Details and frequency of surgery codes (Japanese original codes).**
(DOCX)

**S3 Table. Details and frequency of procedure codes (Japanese original codes).**
(DOCX)

**S4 Table. Details and frequency of drug codes (Anatomical Therapeutic Chemical codes).**
(DOCX)

**S5 Table. Details and distribution of blood-test results.**
(DOCX)

**S6 Table. Hyperparameters in each model for the dataset with the largest number of variables including blood-test results (1543 variables).**
(DOCX)

**S7 Table. The discrimination indicator (c-statistic) of each model for validation.**
(DOCX)

**S8 Table. The goodness of fit indicators of each model for validation.**
(DOCX)

**S9 Table. The calibration indicators of each model for validation.**
(DOCX)

**S10 Table. Regression coefficients in logistic regression with the least absolute shrinkage and selection operator for the data table with the largest number of variables including blood-test results (1543 variables).**
(DOCX)

**S11 Table. The change in the c-statistic by excluding each of the 10 variables with high variable importance in a gradient-boosted decision tree for the data table with the largest number of variables including blood-test results (1,543 variables).**
(DOCX)

## Acknowledgments

We express our gratitude to the personnel of Medical Data Vision Co., Ltd.—particularly Masaki Nakamura, Shogo Atsuzawa, and Masayoshi Suzuki—for their contributions to the preparation of the data. We thank Rina Yamauchi at the School of Public Health, University of Tokyo Graduate School of Medicine for her assistance in creating Supplementary Tables from S1–S4 Tables. We also thank Editage (www.editage.com) for the English language editing.

## Author Contributions

**Conceptualization:** Masao Iwagami.

**Data curation:** Masao Iwagami, Ryota Inokuchi, Nobuaki Michihata, Yuta Taniguchi, Jun Komiyama, Kazuaki Uda.

**Formal analysis:** Masao Iwagami, Ryota Inokuchi.

**Funding acquisition:** Masao Iwagami.

**Methodology:** Ryota Inokuchi, Eiryo Kawakami, Tomohide Yamada, Atsushi Goto, Toshiki Kuno, Yohei Hashimoto, Tadahiro Goto, Tomohiro Shinozaki, Yu Sun, Toshikazu Abe.

**Project administration:** Masao Iwagami.

**Supervision:** Masao Iwagami, Nanako Tamiya.

**Visualization:** Ryota Inokuchi.

**Writing – original draft:** Masao Iwagami.

**Writing – review & editing:** Masao Iwagami, Ryota Inokuchi, Eiryo Kawakami, Tomohide Yamada, Atsushi Goto, Toshiki Kuno, Yohei Hashimoto, Nobuaki Michihata, Tadahiro

Goto, Tomohiro Shinozaki, Yu Sun, Yuta Taniguchi, Jun Komiyama, Kazuaki Uda, Toshi-kazu Abe, Nanako Tamiya.

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
