## [Decision Letter · Decision Letter 0]

27 Mar 2024

PDIG-D-23-00402

Comparison of machine-learning and logistic regression models for prediction of 30-day unplanned readmission in electronic health records: A development and validation study

PLOS Digital Health

Dear Dr. Iwagami,

Thank you for submitting your manuscript to PLOS Digital Health. After careful consideration, we feel that it has merit but does not fully meet PLOS Digital Health's publication criteria as it currently stands. Therefore, we invite you to submit a revised version of the manuscript that addresses the points raised during the review process.

Please submit your revised manuscript within 60 days May 26 2024 11:59PM. If you will need more time than this to complete your revisions, please reply to this message or contact the journal office at digitalhealth@plos.org. Please include the following items when submitting your revised manuscript:

We look forward to receiving your revised manuscript.

Kind regards,

Thomas Penzel

Guest Editor

PLOS Digital Health

Journal Requirements:

Additional Editor Comments (if provided):

Please see the comments in the commented manuscripts and in the detailed sheets from the reviewers. I share their comments and hope for a good revision.

Reviewers' comments:

Reviewer's Responses to Questions

**Comments to the Author**

1. Does this manuscript meet PLOS Digital Health’s publication criteria? Is the manuscript technically sound, and do the data support the conclusions? The manuscript must describe methodologically and ethically rigorous research with conclusions that are appropriately drawn based on the data presented.

Reviewer #1: Yes

Reviewer #2: No

2. Has the statistical analysis been performed appropriately and rigorously?

Reviewer #1: Yes

Reviewer #2: Yes

3. Have the authors made all data underlying the findings in their manuscript fully available (please refer to the Data Availability Statement at the start of the manuscript PDF file)?

Reviewer #1: Yes

Reviewer #2: No

4. Is the manuscript presented in an intelligible fashion and written in standard English?

Reviewer #1: Yes

Reviewer #2: Yes

5. Review Comments to the Author

Reviewer #1: This study compares machine-learning models with logistic regression using electronic health records (EHRs) to predict unplanned readmissions across varying dataset complexities. Some questions need to be addressed properly.

1. The models labeled as "emerging," including gradient-boosted decision trees (GBDT), random forests (RF), and deep neural networks (DNN), are not accurately classified as newly emerging models in the context of machine learning. 

2. The author should consider incorporating an updated review of relevant works from the past five years.

3. It is highly recommended to break down the "Results" section into distinct parts so that the readers can easily navigate and comprehend the findings, facilitating a clearer understanding of the study's outcomes and comparisons.

4. Provide more quantitative metrics or descriptive information to support conclusions. Are there any performance metrics other than C-statistic that would emphasize the advantages of GDBT over LR-LASSO?

5. While the ten most important variables in the GBDT model were identified, more detailed insights should be provided into the contribution of these variables to model performance and how they aid in understanding critical factors related to patient readmission risk.

6. What are the practical implications of the conclusions for actual clinical practice? How could these models be integrated into healthcare systems to improve patient care?

7. The resolution of the images in the article should be improved.

Reviewer #2: 1. Describe the source of data more in details. For instance, are the data maintained by the same system? What type of hospitals are they from? What sizes? What regions?

2. "we intentionally created six pattens of datasets" – do you mean that you simulated data? i.e. generated artificial data? Or you sampled the original data in a certain way? Please elaborate and explain the method used for generating the datasets/

3. Calibration plots (Figure 4) – it would be more typical to arrange them with opposite axes – the predicted (in y axis) vs observed (in x axis)

6. PLOS authors have the option to publish the peer review history of their article (what does this mean?). If published, this will include your full peer review and any attached files.

**Do you want your identity to be public for this peer review?** For information about this choice, including consent withdrawal, please see our Privacy Policy.

Reviewer #1: No

Reviewer #2: No

---

## [Editor Report · Decision Letter 1]

10 Jul 2024

Comparison of machine-learning and logistic regression models for prediction of 30-day unplanned readmission in electronic health records: A development and validation study

PDIG-D-23-00402R1

Dear Dr. Iwagami,

We are pleased to inform you that your manuscript 'Comparison of machine-learning and logistic regression models for prediction of 30-day unplanned readmission in electronic health records: A development and validation study' has been provisionally accepted for publication in PLOS Digital Health.

Best regards,

Thomas Penzel

Guest Editor

PLOS Digital Health

Based on your answers, we are happy to accept.